# Effect of Antibiotic Prophylaxis on Surgical Site Infection in Thyroid and Parathyroid Surgery: A Systematic Review and Meta-Analysis

**DOI:** 10.3390/antibiotics11030290

**Published:** 2022-02-22

**Authors:** Andrea Polistena, Francesco Paolo Prete, Stefano Avenia, Giuseppe Cavallaro, Giovanna Di Meo, Alessandro Pasculli, Fabio Rondelli, Alessandro Sanguinetti, Lucia Ilaria Sgaramella, Nicola Avenia, Mario Testini, Angela Gurrado

**Affiliations:** 1Oncologic and Laparoscopic Surgery, Department of Surgery Pietro Valdoni, Sapienza University of Rome, Policlinico Umberto I, 00161 Roma, Italy; giuseppe.cavallaro@uniroma1.it; 2General and Endocrine Surgery, School of Medical, University of Perugia, S. Maria University Hospital, 05100 Terni, Italy; stefano_avenia@libero.it (S.A.); fabio.rondelli@unipg.it (F.R.); a.sanguinetti@aospterni.it (A.S.); nicola.avenia@unipg.it (N.A.); 3Academic Unit of General Surgery, Department of Biomedical Sciences and Human Oncology, School of Medical, University of Bari Aldo Moro, 70120 Bari, Italy; pretef@gmail.com (F.P.P.); giovanna.dimeo@policlinico.ba.it (G.D.M.); alessandro.pasculli@uniba.it (A.P.); ilaria.sgaramella@policlinico.ba.it (L.I.S.); mario.testini@uniba.it (M.T.); angela.gurrado@uniba.it (A.G.)

**Keywords:** antibiotic prophylaxis, SSI, thyroid, parathyroid, surgery

## Abstract

Thyroid and parathyroid surgery are considered clean procedures, with an incidence of surgical site infection (SSI) after thyroidectomy ranging from 0.09% to 2.9%. International guidelines do not recommend routine antibiotic prophylaxis (AP), while AP seems to be employed commonly in clinical practice. The purpose of this systematic review is analyzing whether the postoperative SSI rate in thyroid and parathyroid surgery is altered by the practice of AP. We searched Pubmed, Scopus, the Cochrane Library, and Web of Science (WOS) for studies comparing AP to no preoperative antibiotics up to October 2021. Data on the SSI rate was evaluated and summarized as relative risks (RR) with 95% confidence intervals (95% CI). Risk of bias of studies were assessed with standard methods. Nine studies (4 RCTs and 5 nRCTs), including 8710 participants, were eligible for quantitative analysis. A meta-analysis showed that the SSI rate was not significantly different between AP and no preoperative antibiotics (SSI rate: 0.6% in AP vs. 2.4% in control group; RR 0.69, 0.43–1.10 95% CI, *p* = 0.13, I2 = 0%). A sensitivity analysis and subgroup analysis on RCTs were consistent with the main findings. Evidence of low quality supports that AP in thyroid and parathyroid surgery produce similar SSI rates as to the absence of perioperative antibiotics.

## 1. Introduction

Thyroid and parathyroid surgery are considered clean head and neck procedures that have no contact with the upper aerodigestive tract (trachea, larynx, pharynx, or esophagus) [1].

With adequate sterilization and advanced operating rooms, postoperative infections are uncommon, and the reported incidence of surgical site infection (SSI) after thyroid surgery is low, ranging from 0.09% to 2.9 [2,3,4,5].

Antimicrobial prophylaxis (AP) was introduced to prevent SSI or an infection related to the operative procedure, which occurs at or near the surgical incision within a 30-day period [1]. SSI determined a prolonged hospital stay and an increased cost for the health system.

International guidelines do not routinely recommend antibiotic prophylaxis, since unnecessary courses are often associated with pathogen resistance and superinfection, potential toxicity, increased cost and hospital stay [6]. Recent guidelines of the American Association of Endocrine Surgeons indicate that AP is not necessary in most cases of standard transcervical thyroid surgery for the management of thyroid disease in adults; however, this recommendation is based on only one randomized controlled trial (RCT) [7], while no mention of AP is made in the guidelines of the American Thyroid Association [8].

On the other hand, AP seems to be employed commonly in clinical practice, often on an open basis, depending on the clinical practice and the behaviors of each center [9,10].

An international survey showed that the rate of antibiotic prophylaxis use varied from 8.8 per cent for European surgeons to 27.9 percent among American surgeons and 58.3 percent by surgeons in Asia [10,11].

In a situation where a divergence appears to exist between guideline recommendations and clinical practice, better information on the efficacy of AP could assist in rationalizing its use and contribute to limit the development of antibiotic resistance. A recent systematic review focusing on SSIs in clean neck surgery did not find evidence in favor of either AP or no AP, while risk of bias was not assessed for the evidence included [12].

Owing to the availability of new studies published on the subject [13,14,15], we conducted a systematic review of the most recent evidence with the aim to guide clinical practice.

The purpose of this systematic review is to analyze whether the incidence of postoperative SSI in thyroid and parathyroid surgery is altered to the practice of AP or not.

## 2. Results

### 2.1. Literature Search and Selection of Studies

The results of our literature search are presented in Figure 1. The initial search identified 1002 results from all databases, with an additional 11 from searching references. After exclusion of duplicates, we screened 760 references and identified 18 eligible references; from these, nine were excluded as not meeting inclusion criteria or not providing data on the outcome of interest [12,16,17,18,19,20,21,22,23]. The remaining nine studies presenting quantitative data were included in the meta-analysis [11,13,14,15,24,25,26,27,28].

### 2.2. Characteristics of Interventions and Populations in the Included Studies

Characteristics of the included studies are presented in Table 1. Four studies were RCTs [14,24,27,28] and five nRCTs [13,15,25,26], published between 2013 and 2021. Two studies were multicentric [11,13], involving 27 and 38 departments of surgery, respectively, while the remaining studies provided data from a single center [13,14,24,25,26,27,28]. The included studies involved 8710 participants (a range of 53 to 2926 per study) from eight countries (Greece, Korea, India, Israel, Italy, Japan, the US and Sweden) in three continents (Europe, Asia and America).

For each study, overall data are presented; split data in the upper row are relative to antibiotic prophylaxis, and in the lower row to the control group (no antibiotic prophylaxis). The mean age ranged from 43.11 to 60.6 years, while the highest prevalence of male sex was 26.6%. Similar inclusion criteria across studies were clean neck surgery for thyroid [11,14,15,25,28] or thyroid and parathyroid diseases [24,26,27,29]. Five studies included only thyroidectomies, while four also included parathyroidectomies; in one study [27] two parotidectomies were also included (3.7%, one per each group). Six studies reported the indication to surgery, which was benign thyroid pathology in two studies [14,28] and a combination of benign and malignant thyroid pathology in the remaining ones. One study included only revision surgery [27]. Five of six studies including malignant pathology [14,15,24,26,27] reported on neck dissections (from a tenth to about a quarter of cases), while the presence of cases of neck dissection was unclear in two studies [11,13]. The mean duration of the procedures was provided by four studies [13,24,25,26], and ranged from 75 to 168 min, while the hospital stay was between three and four days, according to three studies [24,27,28]. Five studies reported the use of a drain [11,13,14,25,28], which varied from 2% [13] to all of the participants [14,25,28], respectively.

Patient skin prepping data were reported by three studies (Table 2) [13,14,24], using either chlorhexidine or povidone iodine solutions. Where data were available, antibiotic prophylaxis consisted of either first, second or third generation cephalosporins [11,13,14,24,26,27,28], administered intravenously, in one to three doses, from 30–60 min before the operation to immediately after intubation [14,24,27,28]. Six studies defined criteria for SSI diagnosis [13,14,15,24,25,27], with four of them [13,24,25,27] using CDC criteria [1], one the Scandinavian Quality Register for Thyroid, Parathyroid and Adrenal Surgery (SQRTPA) criteria [15] and one the Southampton grading scale [14].

#### 2.2.1. RCTs (Cochrane Tool)

Results of the assessment of risk of bias are shown in Figure 2. One of the trials presented limitations for assessment as it was published as conference proceedings and no further data were available [28]. Two trials out of four described the method of randomization used [24,27], which was appropriate in one study [27]. Two trials [24,27] provided information for the assessment of an adequate concealment of allocation procedure, reported the blinding of participants, and produced intention-to-treat analyses (ITT). All studies reported follow-up data. Of the included trials, none had low risk of bias on all items. Two trials [14,28] were of unclear quality in key domains as allocation concealment or blinding of assessors, and three [14,24,27] presented high risk in one of seven domains. Therefore the included trials were deemed to have an overall moderate to high risk of bias.

#### 2.2.2. nRCTs (MINORS Tool)

The MINORS scores are presented in Table 3. All studies included clear aims and outcomes. None of the studies reported a sample size calculation and statistical analysis was most commonly a crosstab statistic.

### 2.3. Primary Outcome

The postoperative rate of SSI was reported in 8710 participants, with an overall rate of SSI of 0.6% in the antibiotic prophylaxis pooled group (0.09–33.3%) and 2.4% in the pooled control group (0–44.3%) (Figure 3); there was no statistically significant difference (RR 0.69, 0.43–1.10 95% CI, *p* = 0.13), and no evidence of heterogeneity for this comparison (I^2^ = 0). The quality of the evidence underpinning this outcome was low.

In order to avoid spurious results due to weak statistical power, the funnel plot and Egger’s regression test were not used due to the small number of retrieved studies (below 10) [30].

A sensitivity analysis was run by excluding, in turn: studies published as conference abstracts, non-randomized studies, revision surgery (as opposed to surgery for primary pathology), surgery for benign thyroid pathology only, studies with unknown follow-up, unclear definition of SSI, unclear antibiotic agent or timing, the only study containing parotid surgery cases (two cases, 3.7% of total cases, one case per study group), and studies with no data on drain positioning, length of stay, operating time, lymphadenectomy. Analysis was also run by the fixed-effect and random-effects models. Results remained consistent in both statistical significance (non-significant difference between SSI among groups) and direction of the evidence.

Subset analysis was conducted on the 4 RCTs included (Figure 4): there were 2380 participants, and no statistically significant difference in SSI between AP and no antibiotics (RR 0.63, 95% CI 0.26–1.53, *p* = 0.31, I^2^ = 0%).

## 3. Discussion

Thyroid and parathyroid surgery are considered clean neck procedures, with an incidence of SSI that in large surveys is less than 1% [3,31]. Many centers report a low incidence of SSIs without the routine use of preoperative AP [2,21], while international guidelines do not suggest routine AP in thyroid and parathyroid surgery [6,8]; but clinical evidence supporting these guidelines for thyroid surgeries is often limited.

Inappropriate use of antibiotics in general increases the possibility of antibiotic resistance and may raise medical costs. However, on the other hand, the occurrence of SSIs can also increase hospital costs, prolong hospital stays and rarely be a cause of death in otherwise healthy patients [32,33]. These reasons often underpin the abuse of antibiotics, and many surgeons still use AP routinely [10,11], with an approach that appears neither supported by current evidence nor influenced by previous experience with SSI after thyroidectomy or parathyroidectomy [10].

The present review contributes to the knowledge on efficacy of AP in thyroid and parathyroid surgery by examining the outcomes of nine studies (4 RCTs and 5 nRCTs) reporting on postoperative SSI in the presence of AP versus no AP. To the best of our knowledge, this is the largest meta-analysis on this topic to date, with 8170 participants included. The overall postoperative incidence of SSIs in thyroid and parathyroid surgery was 1.5% (0.6% in the experimental group and 2.4% among controls). Meta-analysis showed no significant difference in the rate of postoperative SSIs between AP and no preoperative antibiotics administration, with no statistical heterogeneity. These findings do not support the routine use of antibiotic prophylaxis for thyroid surgery, and are in line with recent guidelines observing that the risk of SSIs is not reduced by routine antibiotic prophylaxis in clean neck surgery [6].

Nevertheless, physicians should be aware that severe or even fatal SSIs may develop in specific conditions. Large, non-comparative observational studies have shown that the presence of patient risk factors such as diabetes [21], cardiopulmonary comorbidity [21], ASA score [17] and older age (above 65 or above 80 years) [2,29], disease factors such as the presence of malignancy [34], and procedure-related factors such as prolonged operating time [17,21] and lymphadenectomy [21], may associate with a higher risk of developing post-thyroidectomy SSIs and therefore may benefit from antibiotic prophylaxis. Such evidence has been conflicting in that few of these studies have agreed on the same risk factors for SSIs.

In fact, the development of any infection is multifactorial, and a number of specific factors about an operation’s characteristics may influence the risk of wound infection.

The importance of the sterilization process as an obvious prerequisite to prevent postoperative infection has been highlighted in two of the included studies [13,24], along with that of a periodic independent survey for cleanliness of the operating room and of skin preparation before the surgical procedure [13,17].

Similarly, other factors related to the operative phase were considered associated to the occurrence of SSI, such as the use of absorbable sutures as compared to non-absorbable silk sutures, new surgical devices, and vessel-sealing systems which contribute to decreasing the frequency of ligation, intraoperative blood loss, and operation times [24].

As for the evidence gathered within this systematic review, the included comparative studies found that older age [26], malignancy [11], lymphadenectomy [15], and the use of drains [13,15] was associated to an increased incidence of SSI.

In the present review, however, factors such as lymph node dissection, operation time or rate of drain positioning were not considered in quantitative analysis, primarily by design, but also because not enough studies were available for these outcomes with comparative data for the experimental and control groups.

Because no evidence is provided from this review that may contribute to the comparative evaluation of such risk factors for SSI, the benefits and risks of antibiotic prophylaxis should be considered fully in patients with lymph node dissection and longer operation time, and caution and pre-existing evidence should then be used when deciding on antibiotic prophylaxis in thyroid and parathyroid surgery in the presence of such risk factors.

A sensitivity analysis showed that comparisons that were run only between studies reporting on either lymphadenectomy or drain positioning or operating time did not alter the pooled estimate of effect of SSIs. This seems to be in accordance with a recent large survey conducted in US hospitals where the incidence of SSIs was not influenced by the duration of surgery [31].

A number of limitations should be considered in the present study. Meta-analysis was conducted on a combination of RCTs, portraying either high risk of bias in one key domain or unclear risk in more than one domain, and retrospective observational studies, so the pooled estimate of effect is supported by evidence of low levels. The majority of the included trials portrayed sample sizes, which may suggest statistical underpower in light of the low incidence of SSIs in thyroid and parathyroid surgery [35]. The effect of AP on primary outcomes could not be differentiated by the extent of the intervention, and certain patient characteristics on infection outcomes could not be assessed separately. Another limitation is that data regarding the timing of antibiotics application was incomplete across studies, preventing assessment of effectiveness based on appropriate timing of antibiotic administration. While ample geographic distribution of the studies benefits the external validity of the results of this review, there were no studies from Africa, South America or Australia.

As already stated, the impact of the type of operation and certain patient characteristics on infection outcomes could not be assessed separately.

Despite these limitations, the search strategy of this study is ample and robust and is likely to have identified all of the relevant publications within the inclusion criteria. The number of participants is the highest since the most recent reviews [12,20], and the majority of definitions for SSI and follow-up are consistent and homogeneous. Also, this review is compliant with AMSTAR criteria by design, in contrast to previous systematic reviews.

As a meta-analysis offers a unique opportunity to examine the consistency of definitions and completeness of reporting of data for specific outcomes, we also found in-between study clinical heterogeneity (there was instead no statistical heterogeneity) in the different characteristics of antibiotics for prophylaxis, including administration at different times and for different case-mixes among studies, different definitions for postoperative SSIs and slightly different follow-up times: our analysis suggests the need for defining agreed reporting standards and definitions for key outcomes of efficacy and quality in preoperative AP in thyroid and parathyroid surgery.

Finally, our search returned no studies comparing AP to controls in thyroidectomies performed by remote extra-cervical access. In these patients, there is often an increased duration of surgery, a significant area of dissection and in the particular case of the thyroidectomy by transoral approach with vestibular access, there is a potential contamination of the surgical field with oral bacteria. Further studies are required to assess the usefulness of AP in these settings.

Considering the low quality of the evidence for the primary outcome, with partly different inclusion criteria among studies, differences in postoperative regimens, unclear definition of some of the outcomes, short follow-up and some uncertainty with the estimates of benefits, closely balanced with burdens, the conclusion of this study is that of a substantial equivalence of the two surgical approaches, and further studies could significantly change the results presented herein.

Risks and benefits of antibiotic prophylaxis should be carefully considered in high-risk patients where lymph node dissection and longer operation times are anticipated, or when obesity, diabetes, or other patient or disease high-risk variables are involved. Further research is needed by multi-center studies of adequate methodological quality to provide necessary evidence for the use of AP in thyroid and parathyroid surgery involving high-risk patients.

## 4. Materials and Methods

This review complies with the recommendations of the Cochrane Handbook for Systematic Reviews of Interventions [36], and is reported in line with the preferred reporting items for systematic reviews and meta-analyses (PRISMA) statement [37] and is AMSTAR [38] compliant. Following preliminary searches, a protocol was developed before the review was commenced.

### 4.1. Literature Search

We searched the following databases: the Cochrane Library, Pubmed, Scopus, Web and of Science (WOS) from 1 January 2011 to October 2021.

We used medical subject headings (MeSH) and free-text words. The search strategy (Appendix A) was developed with a professional trial search coordinator to address the following research questions:(Patients) adult patients who underwent thyroidectomy or thyroid lobectomy or parathyroidectomy;(Intervention) preoperative AP;(Comparator interventions) no preoperative AP or placebo;(Outcomes) SSI rate.(Methods-study design) Randomized controlled trials (RCTs) and observational studies. The choice to include both study designs was motivated by the aim to include as much evidence available as possible from existing comparative studies. Observational studies may more frequently enroll larger patient cohorts than RCTs, so given SSIs in thyroid and parathyroid surgery are a relatively rare event, including larger retrospective studies was expected to offer a higher chance to observe a meaningful treatment effect size if data could be pooled in a meta-analysis.

Studies were searched in the English language, in human subjects. There was no restriction of publication status.

The reference lists of all retrieved and relevant publications identified were searched for further studies. The last search was performed on 20 October 2021.

### 4.2. Selection of Studies

The selection of relevant articles was performed in stages. Two independent reviewers (AG, AP) screened the articles retrieved from the initial literature search. Duplicate studies were removed and studies considered irrelevant were discarded. Two reviewers (FPP, AP) further independently reviewed the eligibility of studies in abstract form, or, if appropriate, in full text, by assessing if the inclusion criteria and outcome measures were met. At each stage, reasons for excluding studies were documented. Disagreement regarding article selection was resolved by discussion and consensus or by consulting a third member of the review team (AG). All identified studies were saved in an EndNote X9 database [39]

Each author decided on trial inclusion using predetermined eligibility criteria:-Studies were included if they specifically reported on SSIs by providing numerical data (generic report of postoperative complications or report of no complications without specific mention of SSIs was not considered sufficient for inclusion)-Adult participants (>16 years of age) diagnosed with thyroid and parathyroid diseases undergoing surgery. Study groups including clean neck interventions on organs other than thyroid or parathyroids or lymph nodes were acceptable only if any odd cases were making up for less than 5% of a study population and their prevalence was less than the SSI rate in a study group.-Studies focused on SSI outcomes were included only if data on the proportion of patients receiving antibiotic prophylaxis among SSI cases could be obtained.-Exclusion criteria were:-case reports, technical notes, expert opinions, tutorials, commentaries, protocols with no data, narrative reviews with no original data-comparative studies on clean-contaminated neck surgery-therapeutic, postoperative administration of antibiotic therapy-laboratory studies

### 4.3. Data Extraction

A custom data form, pilot-tested on 10 random studies and approved by agreement between two data abstractors (FPP and AP) was independently used to extract data. Data were recorded onto two Microsoft Excel databases (Version 2019-Windows, Redmond, WA, USA) that were then compared, and any disagreements were reconciled.

In case of publications with partially overlapping data from the same author/institution, the one containing more data or that of higher methodological quality was included.

We extracted data on the following outcomes in all included studies:-study characteristics (authors, publication year, country of origin, study design, sample size, and time interval for each study),-participants’ characteristics (sex, age, inclusion/exclusion criteria and diagnosis),-surgery characteristics (procedure type, operating time in minutes, rate of drain positioning, rate of radical neck dissection),-intervention characteristics (antibiotic, dose, frequency and primary end points), and outcome results (number of events in each group, total infections, length of hospital stay).

Primary outcome was the Rate of Surgical Site Infection (cases/total).

### 4.4. Risk of Bias Assessment

Two researchers (FPP and AP) independently assessed the eligible studies for bias according to the Cochrane Collaboration tool for assessing risk of bias [40] for RCTs and according to the methodological index for non-randomized studies (MINORS) critical appraisal tool for observational studies [41].

For RCTs, seven distinct domains were identified and evaluated as having ‘low risk of bias’ or ‘high risk of bias’ or ’unclear’; sequence generation, allocation concealment, blinding of participants, blinding of outcome assessment, incomplete outcome data, selective outcome reporting and other potential threats to validity.

Non-RCT studies (nRCTs) were assessed for quality and rigor against the MINORS instrument and a global score was assigned to each. The MINORS score is a summation of individual item scores (zero to two for each item), with a maximum of 24 for comparative studies and 16 for noncomparative studies [41].

Disagreement regarding data extraction and quality assessment between reviewers was resolved by consensus or consultation with a third party (AG).

### 4.5. Measures of Treatment Effect

For dichotomous outcomes we extracted the number of patients who experienced the outcome of interest in each group and the number of patients assessed at endpoint in order to estimate a relative risk (RR) and its 95% confidence interval (95% CI).

For continuous outcomes we extracted the final value and standard deviation of the outcome of interest and the number of patients assessed at the endpoint in each treatment arm.

For trials with missing data or in case of doubt, we would attempt to contact the study authors to request data or information to ensure accuracy.

### 4.6. Statistical Analysis

All endpoints were qualitatively summarized. Where clinically similar studies were available, we pooled their results in meta-analyses by using a review manager (Revman version 5.3-Nordic Cochrane Center, Cochrane Collaboration, Odense, Denmark, 2011).

For dichotomous data (e.g., incidence of complications), we used both a fixed and random effects model to calculate a pooled relative risk (RR).

In case of continuous data presented as median and range, we estimated the mean and standard deviation according to the method described by Hozo [42].

Heterogeneity was investigated by the use of the *X*^2^ test and I^2^ statistics. For I^2^ of between 0% and 30%, heterogeneity was considered as probably not important; between 30% and 60%, moderate; between 50% and 90% (or if the *p*-value of *X*^2^ was <0.10), substantial; and between 75% and 100%, considerable [43]. If heterogeneity existed (>30%), we analyzed data using a random effects model. If heterogeneity was not important, a fixed effects model was used. *p* < 0.05 was considered statistically significant. Missing standard deviations were reconstructed from other statistics, such as *p*-values. Possible reasons for substantial heterogeneity were investigated and reported.

We attempted a subgroup analysis considering factors such as study design, sex, age, and type of procedure.

The sensitivity analysis was performed by excluding studies of the lowest quality to explore the degree to which the main findings were affected by the data from individual studies.

## 5. Conclusions

Antibiotic prophylaxis is not associated with the significant prevention of SSI in clean thyroid and parathyroid surgery. The lack of antibiotic prophylaxis is not an independent predictor of infection. This practice is not supported by the available evidence in thyroid and parathyroid surgery, and in absence of a clear indication it should be avoided to reduce costs, adverse reactions, and antibiotic resistance. Antibiotic prophylaxis should only be considered if specific preconditions are present, such as age, an infection diagnosed before surgery, heart/pulmonary comorbidity, diabetes and related increased ASA score, thyroid cancer, or depending on surgical factors such as prolonged operation time, bleeding, extent and duration of surgery and associated lymphadenectomy. Theatre cleaning, sterilization and skin preparation before the surgical procedure are prerequisites to reduce the rate of SSI. The role of drains in otherwise clean thyroid and parathyroid surgery is still debated with regard to their impact on SSI.

## Figures and Tables

**Figure 1 antibiotics-11-00290-f001:**
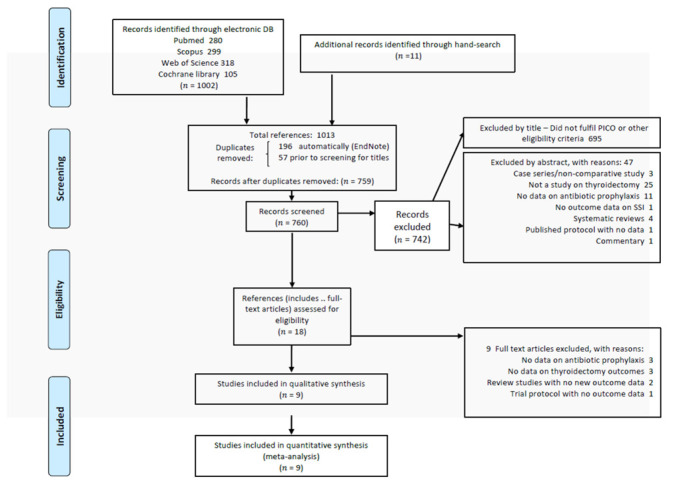
PRISMA flow chart of search results and selection of studies.

**Figure 2 antibiotics-11-00290-f002:**
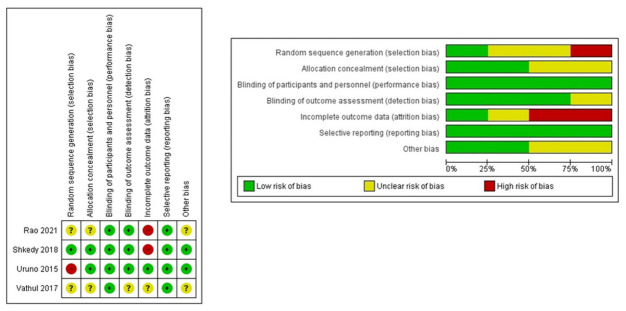
Risk of bias graphic and summary for the included RCTs.

**Figure 3 antibiotics-11-00290-f003:**
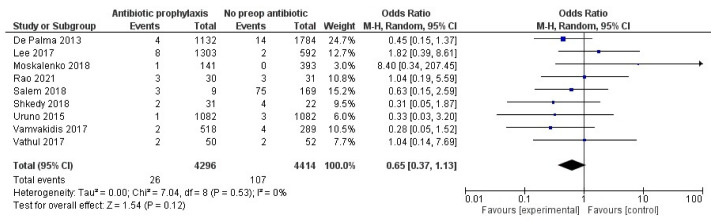
Forest plot for SSI rate in AP group vs. control group (no AP)—All studies.

**Figure 4 antibiotics-11-00290-f004:**
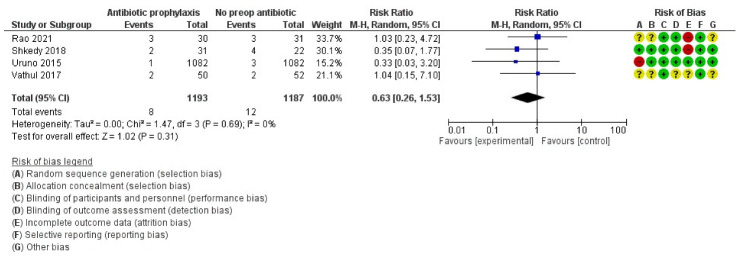
Forest plot for SSI rate in AP group vs. control group (no AP)—RCTs only.

**Table 1 antibiotics-11-00290-t001:** Study design, characteristics of populations, interventions and outcomes in the included studies.

Author, Publication Year, Country	Study Design	Inclusion Criteria	Time Interval	No. Pt	Mean Age, y	Male, %	Procedure	Thyroid Pathology	Radical Neck Dissection, %	Drain, %	Operative Time, min	Lenght of Stay, days	Antibiotic Prophylaxis	SSI, %
De Palma, 2013, Italy	Case control, Multicenter	Thyroid surgery within the study interval	1 January 2009–31 December 2011	2926	52+/−14.6	22.3	TT, n-TT, TL	Benign and malignant (15%-DTC, MTC, anaplastic ca)	NA	91.4	NA	NA	1132	4/1132, 0.35
													1784	14/1784, 0.78
Uruno, 2015, Japan	RCT, Single Center	Clean neck surgery for thyroid or parathyroid disease (Excl.: no consent, sternotomy, resection of trachea, larynx, pharynx, oesophagus, penicillin allergy)	November 2010–April 2012	2164	52+/−15.1	15.6	TT, n-TT, TL, PTX (4%)	Benign and malignant	9	NA	74.7+/−38.1	4	1082 (541 Piperacillin, 541 Cefazolin)	1/1082, 0.09
					52+/−14.7	14.9			8.7		76.1+/−34.0		1082	3/1082, 0.28
Lee, 2017, Korea	Retrospective cohort, Single Center	Thyroid surgery, single Institution, single surgeon (Excl.: endoscopic, robotic surgery)	January 2013–June 2013	1895	44+/−11.4	21.7	TT, lt-TT	Malignant (89% CCND, 11% LND)	12.2	100	108.6+/−56.6	NA	1303	8/1303
			July 2013–December 2013		43.6+/−10.5	26.4			9.6	100	99.0+/−44.0		592	2/592
Vamvakidis, 2017, Greece	Retrospective cohort, Single Center	Clean neck surgery	2010–2014	807	49	20.3	TT, PTX	Benign and malignant (45%-PTC, MTC, other)	8.8	NA	168.5	NA	518	2/518, 0.4
													289	4/289, 1.4
Moskalenko, 2018, USA	Retrospective cohort, Single Center	Thyroid or parathyroid surgery, data from NSQIP database, single center	November 2007–June 2015	534	59.6	23.2	TT, TL, PTX (32.9%)	NA	NA	7/151, 5	79	NA	141	1/141, 0.7
					60.6	21.7				4/393, 1	105		393	0/393, 0
Salem, 2018, Sweden	Nested Case-Control, Multicenter	Thyroid surgery, data from SQRTPA database	2004–2010	218 **	53	26.6	TT, TL	Benign and Malignant	2.5	NA	NA	NA	9	3/9,33.9
						19.2							169	75/16944.3
Shkedy, 2018, Israel	RCT, Single Center	Clean revision H&N surgery, >18 years, no preop indication to AP (Excl.: irradiation, other factors requiring abx, tracheostomy, concurrent infection, penicillin allergy, immunosuppression,)	January 2014–January 2017	53	54.5+/−15.7	19.4	ND, TT, TL, PTX (3.7%), PTD (3.7%)	NA	32.2	NA	NA	3.6+/−1.1	31	2/31, 6.5
					55.5+/−14.2	36.4			13.6			3.5+/−1.1	22	4/22, 18.2
Vathul, 2018, India	RCT, Single Center	Benign (FNAC), TT or TL, >18 <70 years, not Immunocompromised	NA	102	NA	NA	TT, TL	Benign	-	50	NA	4	50	2/50, 4
										52		3	52	2/52, 3.8
Rao, 2021, India	RCT, Single Center	Benign thyroid disease, >16 <80 years, consent (Excl.: diabetes, infective or hematologic disease, other infection, BMI > 25, steroids or immunosuppression, malignancy, drain > 70 mL)	2021	67	44.33+/−7.9	NA	TT, TL	Benign	-	33	NA	NA	33	3/33, 9
					43.11+/−6.9					34			34	3/34, 8.8

For each study overall data are presented; split data in the upper row are relative to antibiotic prophylaxis, in lower row to control group (no antibiotic prophylaxis). NA: data not available; Abx: antibiotics; SSI: Surgical Site Infection; AP: antibiotic prophylaxis; H&N: head and neck; NSQIP: National Surgical Quality Improvement Program; SQRTPA: Scandinavian Quality Register for Thyroid, Parathyroid and Adrenal Surgery; TT: Total Thyroidectomy; n-TT: near Total Thyroidectomy; TL: Thyroid lobectomy: lt-TT: less than TT; CCND: Central Compartment Node Dissection; LND: Lateral Node Dissection; ND: neck dissection; PTX: parathyroidectomy; PTD: parotidectomy; ** 109 SSI and 109 controls matched out of 9494.

**Table 2 antibiotics-11-00290-t002:** Characteristics of antibiotic prophylaxis, definition of SSI and follow-up interval.

Author, Publication Year, Country	Preoperative Patient Skin Prepping	Antibiotic Prophylaxis	Timing	Route	Follow Up	SSI Definition
De Palma, 2013, Italy	NA	cephalosporins or aminopenicillins ± beta lactamase inhibitors	NA	IV	NA	NA
Uruno, 2015, Japan	Chlorhexidine gluconate solution	Piperacillin, 2 g or Cefazolin, 1 g	Immediately after intubation—if operating time > 3 h, further dose	IV	30 days	CDC guidelines for incisional SSI
Lee, 2017, Korea	NA	NA	NA	NA	30 days	CDC guidelines for incisional SSI
Vamvakidis, 2017, Greece	NA	Cefuroxime	NA	IV	NA	NA
Moskalenko, 2018, USA	At the discretion of the operating surgeon (Povidone-iodine used in 96% of patients, or Chloraprep 3.2%)	Cefazolin, Vancomycin, or Clindamycin	NA	IV	30 days	NSQIP criteria (CDC’s definitions for superficial incisional infection, deep incisional infection)
Salem, 2018, Sweden	NA	NA	NA	NA	six weeks	Local wound complication, SQRTPA criteria
Shkedy, 2018, Israel	NA	IV Cefazolin, 1 g (2 g if BMI > 40)	30–60 min before surgery	IV	30 days	CDC guidelines for incisional SSI
Vathul, 2018, India	NA	3rd Gen Cephalosporins	3–4 doses, NA	IV	three months	NA
Rao, 2021, India	Povidone iodine-Betadine for 2–5 min then 0.25 g benzalkonium chloride and 70 g 96% alcohol in concentric circles from the incision site (3 times)	Cefuroxime 1 g	Three doses eight hourly, from induction of anaesthesia	IV	six weeks	Southampton grading scale

CDC: Centers for Disease Control and Prevention; SSI: Surgical Site Infection; NSQIP: National Surgical Quality Improvement Program; SQRTPA: Scandinavian Quality Register for Thyroid, Parathyroid and Adrenal Surgery; IV: intravenous. 2.3. Risk of bias.

**Table 3 antibiotics-11-00290-t003:** Minors scores for comparative studies.

Author, Publication Year, Country	Study Design	Clearly Stated Aim	Inclusion of Consecutive Patients	Prospective Collection of Data	Endpoints Appropriate to the Aim of the Study	Unbiased Assessment of the Study Endpoint	Follow-Up Period Appropriate to the Aim of the Study	Loss to Follow-Up Less Than 5%	Prospective Calculation of the Study Size	An Adequate Control Group	Contemporary Groups	Baseline Equivalence of Groups	Adequate Statistical Analyses	Score
De Palma, 2013, Italy	Case control, Multicenter	2	0	2	2	0	0	2	0	2	2	1 (DRAIN)	2	15
Lee, 2017, Korea	Retrospective cohort, Single Center	2	2	0	2	0	0	2	0	2	0	2	2	14
Vamvakidis, 2017, Greece	Retrospective cohort, Single Center	2	2	0	2	0	2	2	0	2	2	2	2	18
Moskalenko, 2018, USA	Retrospective cohort, Single Center	2	2	2	2	2	2	2	0	2	2	2	2	22
Salem, 2018, Sweden	Matched Case-Control, Multicenter	2	2	2	2	1	2	1	0	2	2	2	2	20

Per-item score: 2 = adequate; 1 = reported but not adequate; 0 = not reported (Overall score: zero = poor, 24 = good). MINORS: methodological index for nonrandomized studies.

## Data Availability

Data are available from the corresponding author upon request.

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
