# Peer review of "Effect of Antibiotic Prophylaxis on Surgical Site Infection in Thyroid and Parathyroid Surgery: A Systematic Review and Meta-Analysis"

_antibiotics, 2022, doi:10.3390/antibiotics11030290_

Round 1

Reviewer 1 Report

The authors present an interesting meta-analysis on the role of antibiotic prophylaxis (AP) in thyroid and parathyroid surgery. The meta-analysis is interesting and well performed and I have only minor comments: have the authors found differentiated data on thyroid and parathyroid surgery? Again, although the difference in SSI is not significant different between patients with AP ant those without, the incidence of SSI is higher in patients without AP: could the authors provide a personal point of view due to they are recognized in this field?

Author Response

R1.The authors present an interesting meta-analysis on the role of antibiotic prophylaxis (AP) in thyroid and parathyroid surgery. The meta-analysis is interesting and well performed and I have only minor comments: have the authors found differentiated data on thyroid and parathyroid surgery? Again, although the difference in SSI is not significant different between patients with AP ant those without, the incidence of SSI is higher in patients without AP: could the authors provide a personal point of view due to they are recognized in this field?

A.Q1.Thank you for your comments. Unfortunately, we could not find distinct data on thyroid and parathyroid surgery when both procedures were presented within each of the included studies. We attributed this to the fact that both thyroid and parathyroid operations were considered as clean neck procedures, so they were included together when comparing postsurgical outcomes between patients who had received antibiotic prophylaxis and patients who did not.

We did not find a specific methodological criterion to pool studies where patients not receiving antibiotic prophylaxis showed higher incidence of postoperative SSIs

Within this systematic review, comparisons indeed showed no significant difference in SSIs between patients who received AP and patients who did not. When examining data we also had the impression that, as a tendency, SSIs appeared to be more frequent in studies where patients who underwent complex or extended procedures did not receive AP.  Although this observation is not supported by the evidence found by this systematic review, this would raise the question whether more complex procedures could be anticipated as situations where  AP could be considered indeed a valid proposition to lower postoperative SSIs. Again, this is pure speculation and is not supported by the evidence from this study. But because other  studies (not included as they did not meet inclusion criteria, in particular they were not comparative studies) highlighted this aspect, their findings were briefly mentioned in the discussion section to comment on our results and provide a context of caution when applying results from this review to the clinical practice.

An English language editing  was performed.

Reviewer 2 Report

There are no clear guidelines regarded the prophylactic use of antibiotics in neck surgery and the clinical practice seams to be quite different.  The authors present a systemic review and metaanalysis on use of antibiotic prophylaxis in neck surgery (thyroid and parathyroid). No significant differences in rate SSI was found between use of antibiotics or controls. The authors present RCTs and nRCTs in separately with similar results. The method and results are clearly presented. 

The conclusion is the use of AP only is patients at high risk or prolonged surgery. This would minimize the use of AP, reduce the cost and mainly reduce the bacterial antibiotic resistance. 

The manuscript is well written, the aims are clear, and the methods are well defined. 

Author Response

R2. There are no clear guidelines regarded the prophylactic use of antibiotics in neck surgery and the clinical practice seams to be quite different.  The authors present a systemic review and metaanalysis on use of antibiotic prophylaxis in neck surgery (thyroid and parathyroid). No significant differences in rate SSI was found between use of antibiotics or controls. The authors present RCTs and nRCTs in separately with similar results. The method and results are clearly presented.

The conclusion is the use of AP only is patients at high risk or prolonged surgery. This would minimize the use of AP, reduce the cost and mainly reduce the bacterial antibiotic resistance.

The manuscript is well written, the aims are clear, and the methods are well defined.

A.R2. Thank you for the time that you took to review in detail this study. An English language editing  was performed.

Reviewer 3 Report

The authors adress a relevant topic, as antibiotics are used way to often in medicine in general. Reviewing several publications regarding antibiotic prophylaxis for surgical site infection in thyroid surgery, they add to the gaining knowledge that antbiotics should be used with restraint in many cases.

However, there are a few points, which need to be discussed. First, there is no differentiation by the extent of the intervention. If this is not possible, it should be mentioned as a limitation. Second, another limitation might be the missing data regarding the time of antbiotics application. Antbiotics should be given some time before cut, which is often not the case in clinical practice. Although, in some cases this might explain the lack of effectivity.

Author Response

R3. The authors adress a relevant topic, as antibiotics are used way to often in medicine in general. Reviewing several publications regarding antibiotic prophylaxis for surgical site infection in thyroid surgery, they add to the gaining knowledge that antbiotics should be used with restraint in many cases.

However, there are a few points, which need to be discussed. First, there is no differentiation by the extent of the intervention. If this is not possible, it should be mentioned as a limitation. Second, another limitation might be the missing data regarding the time of antbiotics application. Antbiotics should be given some time before cut, which is often not the case in clinical practice. Although, in some cases this might explain the lack of effectivity.

A.R3. Thank you for your comments. We fully share your points. Because, on the basis of the evidence included in this study, it was not possible to differentiate outcomes based on the extent of the intervention, and because data regarding the timing of antibiotics application was likewise incomplete or inconsistent across studies,  we included your observations to the limitations of the study and they are presented in the discussion section. An English language editing  was performed.